# The RHO Family GTPases: Mechanisms of Regulation and Signaling

**DOI:** 10.3390/cells10071831

**Published:** 2021-07-20

**Authors:** Niloufar Mosaddeghzadeh, Mohammad Reza Ahmadian

**Affiliations:** Institute of Biochemistry and Molecular Biology II, Medical Faculty of the Heinrich Heine University, Universitätsstrasse 1, Building 22.03.05, 40225 Düsseldorf, Germany; mosaddeg@uni-duesseldorf.de

**Keywords:** CDC42, effectors, RAC1, RHOA, RHOGAP, RHOGDI, RHOGEF, RHO signaling

## Abstract

Much progress has been made toward deciphering RHO GTPase functions, and many studies have convincingly demonstrated that altered signal transduction through RHO GTPases is a recurring theme in the progression of human malignancies. It seems that 20 canonical RHO GTPases are likely regulated by three GDIs, 85 GEFs, and 66 GAPs, and eventually interact with >70 downstream effectors. A recurring theme is the challenge in understanding the molecular determinants of the specificity of these four classes of interacting proteins that, irrespective of their functions, bind to common sites on the surface of RHO GTPases. Identified and structurally verified hotspots as functional determinants specific to RHO GTPase regulation by GDIs, GEFs, and GAPs as well as signaling through effectors are presented, and challenges and future perspectives are discussed.

## 1. Introduction

The RHO (RAS homolog) family is an integral part of the RAS superfamily of guanine nucleotide-binding proteins. RHO family proteins are crucial for several reasons: (i) approximately 1% of the human genome encodes proteins that either regulate or are regulated by direct interaction with RHO proteins; (ii) they control almost all fundamental cellular processes in eukaryotes including morphogenesis, polarity, movement, cell division, gene expression, and cytoskeleton reorganization [1]; and (iii) they are associated with a series of human diseases (Figure 1) [2].

The RHO family of proteins functions as molecular switches in the cell and cycle between being in a GDP-bound, inactive state and a GTP-bound, active state [3]. Invaluable insights have been gained by structural and biochemical studies of RHO GTPases and their complexes with interacting partners thus far, increasing our understanding of both how the switch mechanism of the RHO GTPases is regulated and how a RHO GTPase can interact with four classes of structurally and functionally unrelated protein families (Figure 1) [3]. The cellular regulation of this cycle involves guanine nucleotide exchange factors (GEFs), which accelerate intrinsic GDP/GTP exchange, and GTPase-activating proteins (GAPs), which stimulate intrinsic GTP hydrolysis activity [4]. The formation of the active GTP-bound state of the GTPase is accompanied by a conformational change in two regions (known as switch I and II), which provides a platform for the selective interaction with structurally and functionally diverse proteins (the so-called downstream effectors; Table 1) that initiate a network of cytoplasmic and nuclear signaling cascades [5,6]. A prerequisite of RHO protein function is membrane association, which is achieved by isoprenylation, a posttranslational modification. In this respect, RHO proteins are regulated by a third control mechanism that directs their membrane targeting to specific subcellular sites. Specifically, guanine nucleotide dissociation inhibitors (GDIs) bind selectively to prenylated RHO proteins and control their cycle between the cytosol and membrane. Activation of RHO proteins results in their association with effector molecules that subsequently activate a wide variety of downstream signaling cascades, thereby regulating many important physiological and pathophysiological processes in eukaryotic cells [7].

The molecular mechanisms of RHO GTPase regulation have been well characterized, but our understanding of the signal transduction to downstream targets and, most notably, the autoinhibitory mechanisms of GEFs, GAPs, and effectors remain unclear. Very important and challenging, the elucidation of these critical control mechanisms will open new directions for the design of additional therapeutic interventions.

Signaling by these GTPases is controlled by other mechanisms including post-translational modifications such as phosphorylation, ubiquitylation, sumoylation, and acetylation (see for more details [8,9]).

**Table 1 cells-10-01831-t001:** RHO GTPases, potential effectors and their functions in mammalian cells.

RHO GTPases	Effector Proteins	Function	Functions and Effects	References
RHOA	ROCK I/II	Ser /Thr kinase	Actin myosin contraction, Stress fiber formation	[10,11]
	Citron kinase	Ser /Thr kinase	Cytokinesis	[12]
	MBS	Phosphatase subunit	MLC inactivation	[13]
	DIA 1/2	Formin-like proteins	Actin polymerization	[10]
RHOB	Integrin β1	Cell surface receptor	Cell adhesion and migration	[14]
RHOC	FMNL3	Formin like proteins	Migration, Invasion	[15]
RHOH	Kaiso	Transcription factor	TCR activation	[16]
RAC1	PAK1/2/3 *	Ser /Thr kinase	JNK activation, Actin filament stabilization	[17]
	MLK 2/3 *	Ser /Thr kinase	JNK activation	[18,19]
	WAVE	Scaffold	Actin organization	[20]
	p70 S6 kinase *	Ser /Thr kinase	Translation regulation	[21]
	IQGAP1/2 *	Scaffold	Actin/cell-cell contacts	[22,23]
	MEKK1/4 *	Ser /Thr kinase	JNK activation	[24]
	POR1	Scaffold	Actin organization	[25]
	p67^phox^*	Scaffold	ROS generation	[26]
	PI3 kinase	Lipid kinase	PIP3 levels	[27]
	DAG kinase	Lipid kinase	PA levels	[28,29]
	PLCβ2 *	Lipase	DAG and IP3 levels	[30]
RAC1B	p120^ctn^	Catenin	Cellular transformation	[31]
RAC2	LFA-1	Cell surface receptor	B cell adhesion	[32]
RAC3	GIT1	ARF GAP and scaffold	Regulation of cell adhesion and differentiation	[33]
RHOG	Kinectin	Kinesin receptor	Microtubule dependent transport	[34]
CDC42	N-WASP	Scaffold	Actin organization	[35]
	PAK4	Ser/Thr kinase	Actin organization	[36]
	MRCKα/β	Ser/Thr kinase	Actin organization	[37]
TCL	GIT-PIX complex	Scaffold	Stabilization of focal adhesion	[38,39]
RHOD	Plexin A1/B1	Semaphorin co-receptor	Growth cone formation	[40]
RIF	DIA 1/2	Formin-like proteins	Actin organization	[40,41]
RND1	Stathmin2	Neuronal growth associated proteins	Microtubule depolymerization, Neurite extension	[42]
RND2	Rapostlin	Formin-binding protein	Neurite branching	[43]
RND3	Socius	Scaffold	Loss of stress fibers	[44]
	ROCKI	Ser/Thr kinase	Actomyosin contractility	[45,46]

* Proteins shown with an asterisk are shared effectors for both RAC1 and CDC42.

## 2. The RHO Family and the Molecular Switch Mechanism

Members of the RHO family have emerged as key regulatory molecules that couple changes in the extracellular environment to intracellular signal transduction pathways. To date, 20 canonical members of the RHO family have been identified in humans and can be categorized into distinct subfamilies based on their sequence homology: RHO (RHOA, RHOB, and RHOC); RAC (RAC1, RAC1B, RAC2, RAC3, and RHOG); CDC42 (CDC42, G25K, TC10, TCL, WRCH1, and WRCH2); RHOD (RHOD, RIF); RND (RND1, RND2, and RND3); and RHOH [47].

RHO family proteins are approximately 21–25 kDa in size. They typically contain a conserved GDP-/GTP-binding domain, called the G domain, and a C-terminal hypervariable region (HVR) ending with a consensus sequence known as CAAX (C is cysteine, A is any aliphatic amino acid, and X is any amino acid) (Figure 2). The G domain consists of five conserved sequence motifs (G1 to G5) that are involved in nucleotide binding and hydrolysis [48]. In the cycle between the inactive and active states, at least two regions of the protein, switch I (G2) and switch II (G3), undergo structural rearrangement and transmit an “OFF” to “ON” signal [3]. Subcellular localization, which is known to be critical for the biological activity of RHO proteins, is achieved through a series of posttranslational modifications at a cysteine residue in the CAAX motif including isoprenylation (geranylgeranyl or farnesyl), endoproteolysis, and carboxyl methylation [49].

**Figure 2 cells-10-01831-f002:**
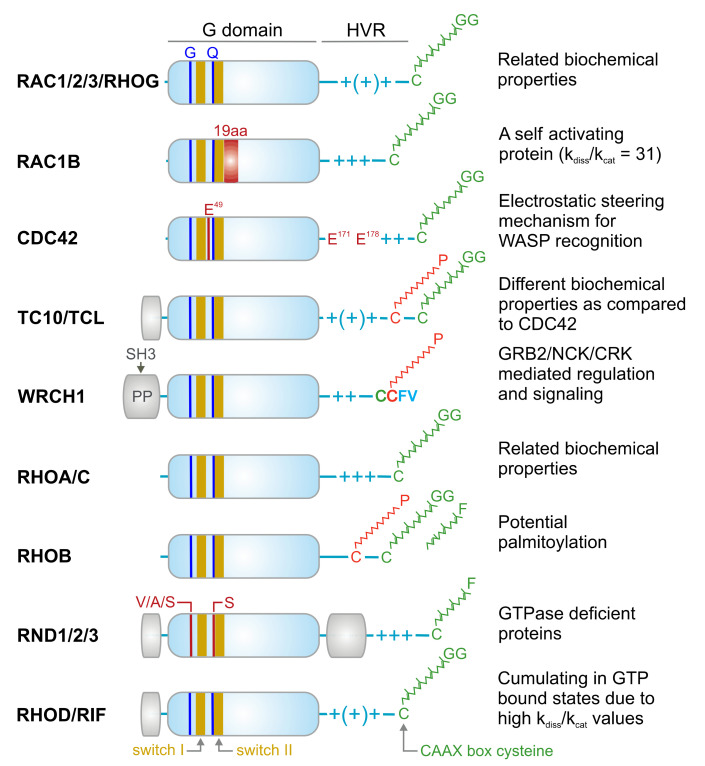
Domains, signature motifs, and post-translational modification of RHO GTPases. RHO GTPases contain a highly conserved G domain, which is responsible for GDP/GTP binding and GTP hydrolysis. Switch I and switch II regions are the consensus binding sites for GEFs, GAPs, GDIs, and effectors, and undergo conformational changes upon the nucleotide exchange and hydrolysis [3]. All members of the RHO family contain conserved glycine 12 (G) and glutamine 61 (Q; RAC1 numbering), except for the RND proteins, which contain, among other deviations, other residues at these positions. This is why RND proteins constantly remain in the GTP bound state [50]. Other signatures are, for example, a 19-amino acid insertion next to the switch II region in RAC1B with drastic biochemical consequences [51], and glutamic acids (E) in CDC42 crucial for a selective WASP interaction [52]. Some members have amino acid insertion outside the G domain (yellow boxes) with yet unknown properties. The N-terminal insertion in WRCH1 contains proline-rich motifs responsible for interaction with SH3-contining adaptor proteins [53]. Most members have comparable biochemical properties such as nucleotide binding, exchange, and hydrolysis. In contrast to most members, which end up under resting conditions in an inactive GDP-bound state, RAC1B, RHOD, and RIF cumulate in the GTP-bound state due to a faster intrinsic nucleotide exchange reaction (k_dis_) compared to the intrinsic GTP hydrolysis reaction (k_cat_) [54]. The C-terminal hypervariable region (HVR) contains the terminal CAAX box, which undergoes posttranslational modification by geranylgeranylation (GG) or alternatively farnesylation (F) in the case of RHOB and the RND proteins at the conserved cysteine (green). Additional modification by a palmitoyl (P) moiety has been reported for RHOB, and the CDC42-related proteins TC10, TCL, and WRCH1. These modifications lead to the membrane anchorage of the members, a process that is stabilized and potentiated through variable numbers of positively charged arginine and lysine residues (+).

Once an isoprenoid moiety is added to CAAX, a RHO protein is translocated to the endoplasmic reticulum, where RCE1 cleaves the AAX tripeptide tail, and then, RHO undergoes carboxymethylation by ICMT [55] RHO proteins can also be phosphorylated, which can affect their association with their regulators or effectors or influence their membrane stability [56,57,58].

A characteristic region of RHO family GTPases is the insert helix (amino acids 124–136, RHOA numbering), which may play a role in effector activation and downstream processes [59].

Although the majority of the RHO family proteins are remarkably inefficient GTP-hydrolyzing enzymes, in quiescent cells, they accumulate in an inactive state because GTP hydrolysis by RHO proteins is, on average, two orders of magnitude faster than GDP/GTP exchange [47]. These different intrinsic activities provide the basis for a two-state molecular switch mechanism, which greatly depends on the regulatory functions of GEFs and GAPs. Eleven of the 20 RHO family members possess classical molecular switches, namely, RHOA, RHOB, RHOC, RAC1, RAC2, RAC3, RHOG, CDC42, G25K, TC10, and TCL [47].

Atypical RHO family members including RND1, RND2, RND3, RAC1B, RHOH, WRCH1, RHOD, and RIF have been proposed to accumulate in the GTP-bound form in cells [47]. RND1, RND2, RND3, and RHOH constitute a completely distinct group of proteins within the RHO family (Figure 2) [60], as they do not share several essential amino acids including Gly-12 (RAC1 numbering) in the G1 motif (a phosphate-binding loop or P-loop) and Gln-61 in the G3 motif or switch II region, which are critical in GTP hydrolysis. Thus, they can be considered GTPase-deficient RHO-related GTP-binding proteins [61]. RHOD and RIF are involved in the regulation of actin dynamics [41] and exhibit much faster nucleotide exchange than GTP hydrolysis. WRCH1, a CDC42-like protein that has been reported to be a fast-cycling protein, resembles RAC1B, RHOD, and RIF in this regard (Figure 2) [47]. These atypical members do not possess the classical switch mechanism and, therefore, may be regulated through other mechanisms.

## 3. Regulation of RHO Family GTPases

### 3.1. Guanine Nucleotide Dissociation Inhibitors (GDIs)

Despite the vast number of RHOGEFs and RHOGAPs, only three GDIs exist in the human genome. The RHOGDI family includes ubiquitously expressed GDI1 (or RHOGDIα) [62]; GDI2 (GDIβ, LY-GDI or D4-GDI), mainly in hematopoietic tissue [63]; and GDI3 (or GDIγ), which is usually expressed in human cerebral, lung, and pancreatic tissue [64]. An N-terminal extension that anchors GDI3 to the membrane of Golgi vesicles distinguishes this isoform from the others [65].

Several studies in recent decades have provided information about the structure and function of GDIs and proposed that they act as shuttles for RHO GTPase [8,66,67,68]. The shuttling process is initiated by the release of RHO GTPases from donor membranes, the formation of inhibitory cytosolic GDI-RHO GTPase complexes, and the delivery of RHO GTPases to the membranes of subcellular compartments [66,67].

It has been demonstrated that the isoprenylation process in cells can be regulated by GDIs [69]. GDI mediates the release of RHO GTPases from the membrane, maintains them in an inactivated state, and safeguards them against degradation or nonspecific activation by RHOGEFs [25,29,30]. Different structural studies have revealed two sites of GDI and RHO GTPase interaction [70,71,72,73,74]. First, an N-terminal regulatory arm of GDI binds to the switch region of RHO GTPases and inhibits GDP dissociation and GTP hydrolysis. Second, the N-terminus of GDI attracts the positively charged RHO hypervariable region, which is engaged with negatively charged phospholipids of the membrane and initiates the insertion of the geranylgeranyl moiety on the RHO GTPases into a hydrophobic pocket in the GDI molecule, leading to membrane release [75].

### 3.2. Guanine Nucleotide Exchange Factors (GEFs)

GEFs are able to selectively bind to their respective RHO proteins and accelerate the exchange of tightly bound GDP for GTP [8]. Typically, GEFs profoundly reduce the affinity of RHO proteins for GDP, leading to its displacement from GDP and subsequent association with GTP [76,77]. This reaction involves several stages including an intermediate state in which the GEF is in the complex with the nucleotide-free RHO protein. This intermediate does not accumulate in the cell and rapidly dissociates because of the high intracellular GTP concentration, leading to the formation of the active RHO-GTP complex. The main principle driving this mechanism is based on the binding affinity of nucleotide-free RHO protein being significantly greater for GTP than for GEF proteins [76,78]. Cellular activation of RHO proteins and their cellular signaling can be selectively uncoupled from GEFs through the overexpression of dominant-negative mutants of RHO proteins (e.g., threonine 19 in RHOA is replaced with asparagine) [79]. Dominant-negative mutants form a tight complex with their cognate GEFs, preventing them from activating endogenous RHO proteins. RHOGEFs are classified into two distinct families: DBL homology (DH) domain-containing proteins, and dedicator of cytokinesis (DOCK) proteins [80,81].

#### 3.2.1. DBL Family GEFs

RHOGEFs of the diffuse B-cell lymphoma (DBL) family directly activate the proteins of the RHO family [82]. The prototype of this GEF family is the DBL protein, which was isolated as an oncogenic product from diffuse B-cell lymphoma cells in an oncogene screen [83] and was later reported to act on CDC42 [84]. Human DBL family proteins have recently been grouped into functionally distinct categories based on both their catalytic efficiencies and their sequence–structure relationship [47]. Members of the DBL family are characterized by a unique DBL homology (DH) domain [85,86,87,88].

The DH domain is a highly efficient catalytic machine [80] that is able to accelerate the nucleotide exchange of RHO proteins by as much as 10^7^-fold. The DH domain is often followed by a pleckstrin homology (PH) domain, indicating its essential and conserved function. A model for PH domain-assisted nucleotide exchange has been proposed for some GEFs such as DBL, DBS, and TRIO [80]. Thus, the PH domain serves multiple roles in signaling events by anchoring GEFs to the membrane (via phosphoinositides) and directing them toward their respective GTPase partners, which are on the membrane [80].

Through a search for DH domain-containing proteins in the human genome, 74 DBL proteins have been identified (Figure 3) [47]. Interestingly, nine of these DBL proteins lack the C-terminal tandem PH domain, and three of these proteins contain a membrane bending and tubulating BAR (BIN/amphiphysin/RVS) domain, and seven of 20 investigated DBL proteins do not exhibit any GEF activity (Figure 3) [47]. In addition to the DH-PH tandem motif, DBL family proteins are highly diverse and contain additional domains with different functions (Figure 3) including SH2, SH3, CH, RGS, PDZ, and/or IQ domains, which enable their interaction with other proteins; BAR, PH FYVE, C1, and C2 domains, which enable their interaction with membrane lipids; and other functional domains such as Ser/Thr kinase, RASGEF, RHOGAP, and RANGEF [82]. These additional domains have been implicated in autoregulation, subcellular localization, and connection to upstream signaling molecules [40,49,50]. Spatiotemporal regulation of DBL proteins has been suggested as a mechanism that specifically initiates the activation of substrate RHO proteins and controls a broad spectrum of normal and pathological cellular functions [89]. Thus, it is evident that members of the DBL protein family are attractive therapeutic targets for a variety of diseases [90,91].

#### 3.2.2. Structural and Functional Characteristics of the DH domain

The DH domain is the signature of DBL family proteins. The catalytic guanine nucleotide exchange activity of DBL family proteins is realized entirely within the DH domain, which is not only sufficient for catalytic activity but also critical for substrate specificity [47,92]. The catalytic DH domain consists of approximately 200 residues, and as determined by x-ray and NMR analyses of the DH domain in several DBL proteins, it is composed of a unique extended bundle of 10–15 alpha helices [93]. This helical fold is mainly composed of three conserved regions, CR1, CR2, and CR3, each of which is 10–30 residues long and forms separate alpha helices that are packed together [45,53]. The CR1 and CR3 regions are solvent exposed until complexed with RHO proteins [47]. Except for these three conserved regions (CR1, CR2, and CR3) in DH domains, DBL family members share little homology with each other [87].

#### 3.2.3. The Tandem PH Domain in DBL Proteins

In the majority of DBL family proteins, the catalytic DH domain is followed by a PH domain consisting of approximately 100 residues (Figure 3), and even though the identity of the PH domain among members of the DBL family is less than 20%, the PH-domain containing DBL proteins share a similar three-dimensional structure with two orthogonal antiparallel β-sheets and a folded C-terminal α-helix that cover one end [94,95]. The PH domain was originally identified in a number of cytoplasmic signaling proteins that displayed homology with a region repeated in pleckstrin [96,97]. The DH-PH tandem is a signature motif of the DBL family, indicating that the PH domain has an essential and conserved function [85,88]. The tandem PH domain can act as a “membrane-targeting device” due to its ability to bind phosphoinositides [98]. It can also bind directly to RHO proteins and potentiate the DH-catalyzed nucleotide exchange reaction [92,93]. In contrast, the PH domains have been shown to bind and inhibit the activity of the DH domain [99,100]. In addition to its membrane-targeting properties, emerging evidence suggests that the PH domain may also play important regulatory roles by serving as a protein–protein interaction module [101].

#### 3.2.4. A Plethora of DBL Family Proteins

It is evident that DBL family proteins are more abundant and varied in cells than RHO family proteins. To date, 74 DBL proteins have been reported in humans, and they are classified into different subfamilies: 46 DBL proteins are monospecific for RHO-, RAC-, and CDC42-selective proteins, five are bispecific for RHO- and CDC42-selective proteins, and six are oligospecific for all three RHO protein subgroups [47]. Since there are many more DBL proteins and many of them can activate more than one RHO protein, the activation of RHO proteins catalyzed by DBL family proteins constitutes a level of regulation in which the signaling pathways can converge or diverge toward one or more RHO proteins [7]. This multifunctionality suggests that at least one representative of each DBL subfamily is expressed in all mammalian cells, but they may act at distinct subcellular sites.

### 3.3. DOCK Family of RHOGEFs

The 11 members of the DOCK family can be categorized into four subfamilies: DOCK-A, DOCK-B, DOCK-C, and DOCK-D [81,102]. DOCK GEFs have two conserved domains: lipid-binding DOCK homology region 1 (DHR-1), which facilitates DOCK localization to membrane compartments, and catalytic DOCK homology region 2 (DHR-2), which induces the GDP-GTP exchange reaction [81,102,103,104]. It has been proposed that DOCK GEFs activate RAC1 and CDC42 proteins, but not other RHO proteins [105,106].

DOCK proteins orchestrate important processes in brain development including neuron, microglial, and Schwann cell development and functions [102,107]. DOCK2 and DOCK8 play significant roles in immune responses such as the chemotactic responses of T cells and B cells, ROS production in neutrophils, and migration of mature dendritic cells [81]. Li et al. demonstrated that DOCK1 forms a complex with ELMO1, RAC1, RAC2, and Gαi2, which initiates actin polymerization in breast cancer cells [108].

DOCK2 has been indicated to increase amyloid beta plaque formation, which makes this protein a potential Alzheimer’s therapeutic target [109,110]. Janssen et al. showed that in T cells, DOCK8 can form a complex with WASP and ARP2/3 and link TCR to the actin cytoskeleton to form a synapse for T cell responses [111].

Overall, DOCK GEFs play pivotal roles in different biological processes that can be dependent or independent of their GEF activity.

### 3.4. GTPase-Activating Proteins (GAPs)

Hydrolysis of bound GTP is the timing mechanism that terminates signal transduction of RHO family proteins and enables their return to an inactive, GDP-bound state [87]. The intrinsic GTPase reaction is usually slow but can be stimulated to accelerate by several orders of magnitude through interaction with RHO-specific GAPs [112,113]. The RHOGAP family is identified by the presence of a conserved catalytic GAP domain that is sufficient for engaging RHO proteins and mediating accelerated catalysis [114,115]. The GAP domain inserts a conserved arginine residue, termed an “arginine finger”, into the GTP-binding site of the cognate RHO protein to stabilize the transition state and catalyze the GTPase reaction [74,76,77]. This mechanism is similar to that of other small GTP-binding proteins including RAS, RAB, and ARF, although the sequence and folding of the respective GAP families differ from other GTP-binding proteins [115,116]. Masking the catalytic arginine finger is an elegant mechanism for the inhibition of GAP activity. This action has also been recently discovered in the tumor suppressor protein DLC1, an RHOGAP, which is competitively and selectively inhibited by the SH3 domain in p120RASGAP [117,118].

The first RHOGAP discovered, p50RHOGAP, was identified through a biochemical analysis of human spleen cell extracts in the presence of recombinant RHOA [119]. The majority of RHOGAP family members typically harbor several other functional domains and motifs that are implicated in tight regulation and membrane targeting (Figure 4) [74,82,83]. Numerous mechanisms have been shown to affect the specificity and catalytic activity of RHOGAPs (e.g., intramolecular autoinhibition [120], posttranslational modification [121], and regulation by interaction with lipid membranes [122] and proteins [118]).

RHOGAP insensitivity has been frequently analyzed through the substitution of either amino acid that is critical for GTP hydrolysis by RHO proteins (e.g., Gly14 or Gln63 in RHOA), and these mutations generate proteins known as constitutively active mutants [123,124]. In other mutants, the catalytic arginine residue of the GAP domain is replaced with an alanine residue [113,124]. The latter approach is, in principle, very useful under cell-free conditions but not optimal in cells because an Arg-to-Ala mutant may provide a readout similar to that of the wild-type protein as it interferes with downstream signaling by competing with effector(s) for binding to RHO proteins. These RHOGAP mutants are able to bind persistently to their target protein, sequestering the target, which most likely leads to a readout similar to that of activated wild-type RHOGAP. Therefore, it has recently been suggested that mutating critical “binding determinants”, particularly Lys319 and Arg323 (p50 numbering), may be a better strategy than substituting the catalytic arginine [114]. Charge reversal of these residues most likely leads to loss of RHOGAP association with its substrate RHO protein and thus abrogates the activity of the GAP domain. This outcome renders mutagenesis not only a tool for determining the specificity of RHOGAPs, but also for investigating GAP domain-independent function(s) of the RHOGAPs.

#### 3.4.1. RHOGAP Family Proteins

The GTPase reaction is of great medical significance, since any disruption of this reaction such as that caused by inhibitory mutations in genes encoding GAP proteins results in persistent downstream signaling. The discovery that GAPs are required for GTPase downregulation was made on the basis that microinjection of recombinant GTP-bound RAS into living cells results in faster GTP hydrolysis than is realized in vitro [125]. The first discovered RHOGAP, p50RHOGAP, was identified by biochemical analysis of human spleen cell extracts with recombinant RHOA [119], and this discovery led to the identification of other RHOGAP-containing proteins such as chimaerin and BCR, whose amino acid sequences are related to p50RHOGAP [126]. Since then, more than 66 RHOGAP-containing proteins have been identified in humans [114,127] The RHOGAP family is identified by the presence of a conserved catalytic GAP domain that is sufficient for RHOGAP interaction with RHO proteins and, in most cases, stimulation of the intrinsic GTP hydrolysis reaction of RHO GTPases [115]. In addition to their signature RHOGAP domain, most RHOGAP family members frequently harbor several other functional domains (Figure 4). The majority of these domains can be classified into the following three major groups: (i) lipid- and membrane-binding domains; (ii) peptide- and protein-interacting domains; and (iii) catalytic domains with enzymatic activities. The most widespread domains are PH, CC, P, SRC homology 3, and BAR/F-BAR (Figure 4). These domains are implicated in regulation, membrane targeting, localization, and potential phosphorylation sites and indicate the complexity of the regulation of GTPase activity. Thirteen GAPs lack any additional putative domains but contain highly variable regions in their N- and C-termini (Figure 4). It is possible that these regions consist of motifs that have not yet been identified, and these regions may contribute to their specific function in the cell.

#### 3.4.2. Structural and Functional Characteristics of the RHOGAP Domain

The GAP domain of the RHOGAP family consists of approximately 190 amino acids and shares high sequence homology within the family. Although the RHOGAP domain shares no similarities to RASGAP family members at the amino acid level, RHOGAPs and RASGAPs resemble each other in their tertiary structure [128,129]. Comparative structural analysis of the RHOGAP domain with other GAPs of RAS subfamilies has suggested that GAP domains in RAS and RHO family proteins are evolutionarily related [128,130] and that the catalytic domains of RHOGAPs share a core structural fold. The RHOGAP domain is made up of seven α-helices. The functional characteristic of the RHOGAP domain is a pair of conserved basic residues: catalytic arginine (the arginine finger) and lysine (Arg282 and Lys319 in p50RHOGAP numbering) [114,131].

#### 3.4.3. The Mechanism by Which the GAP Domain Mediates GTP Hydrolysis

Crystallographic studies of RHOGAP domains in complex with CDC42 bound to GppNHp, RHOA/CDC42 bound to GDP·AlF_4_ [76,77,95] and RHOA bound to GDP·MgF_3_ [132] have provided insights into the catalytic mechanism of GTP hydrolysis upon stimulation. The GTPase reaction, as part of the switch mechanism, leads to changes in the conformation of the GTPase, especially in flexible and mobile loops known as switch regions. RHOGAP interacts with the switch I and II regions [3,133] and the P-loop of the RHO protein. The GAP domain accelerates the intrinsic GTP hydrolysis by RHO proteins in two ways. First, it directly contributes to catalysis by inserting catalytic arginine in the GAP domain into the active site of the RHO protein. This establishes contacts with the main-chain carbonyl of Gly12 (RAC1 numbering) and helps stabilize the GTP-hydrolysis transition state [134]. Second, this interaction stabilizes the negative charges formed during the transition state of GTP hydrolysis and positions the catalytic glutamine residue (Gln61 RAC1 numbering) of the RHO protein to enable its coordination with nucleophilic water molecules [129,135]. RHOGAP also stabilizes the switch regions of the RHO protein by interacting with residues associated with its intrinsic GTPase activity [113]. ARHGAP36, CNTD1, DEP1, DEP2, FAM13B, INPP5P [136], and OCRL1 lack an arginine finger, which makes them catalytically inactive (Figure 4) [114]. ARHGAP36 is involved in GLI transcription factor activation, but this function proceeds independent of its GAP domain. CNTD1 lacks RHOGAP activity and acts as an ARF6 GAP. DEP1 and DEP2 coordinate cell cycle progression and interfere with RHOA action and signaling even though they lack RHOGAP activity. OCRL1 has been shown to interact with GTP-bound RAC1 without stimulating hydrolysis. p85α and p85β (85-kDa regulatory subunits of phosphoinositide 3-kinases) are also RHOGAP-like proteins (Figure 4), as they do not show any detectable GAP activity toward different RHO proteins [28]. A prerequisite of GAP function is that the GAP domain position its catalytic residue Arg282 (p50 numbering); therefore, GAPs include a number of amino acids that are critical for binding and stabilizing the protein complex. Both p85 isoforms lack most of binding determinants (e.g., Arg323, Asn391, Val394, and Pro398) as well as the conserved amino acids around the arginine finger [114].

#### 3.4.4. Overabundance and Diversity

Using database searches, 66 distinct RHOGAP domain-containing proteins were found to be encoded in the human genome, whereas the number of RHO family proteins that need to be regulated by GAPs was 18 (excluding constitutively active RHO proteins). The overabundance of RHOGAPs implies that they must be tightly regulated in the cell to prevent RHO proteins from being accidentally turned off. Of the 66 RHOGAPs, 57 proteins have a common catalytic domain capable of terminating RHO protein signaling by stimulating the slow intrinsic GTP hydrolysis (GTPase) reaction (Figure 4). Investigation of the sequence-structure-function relationship between RHOGAPs and RHO proteins by combining in vitro data with in silico data has revealed that the RHOGAP domain itself is nonselective, and in some cases, it is rather inefficient under cell-free conditions. This finding suggests that other domains in RHOGAPs confer substrate specificity and fine-tunes their catalytic efficiency in cells [114].

#### 3.4.5. Regulation and GAP Proteins Functions

RHOGAPs are widely expressed, which makes their apparent redundancy questionable. Therefore, cells must regulate RHOGAPs very tightly to prevent unwanted events that switch off signaling. To ensure stringent regulatory control, RHOGAPs are modulated at different levels, indicating that regions outside the RHOGAP domain most likely determine the specificity of RHOGAPs (Figure 4). Numerous mechanisms have been shown to affect the catalytic activity and substrate specificity of RHOGAPs (e.g., autoinhibition (GRAF and OPHN1) [120]); posttranslational regulation such as phosphorylation (p190GAP and Mgc-RACGAP) [121]; lipid binding via PH or C2 domains [122]; protein–protein interactions (DLC1/p120RASGAP) [117,118] and subcellular distribution through specific colocalization of RHOGAPs with RHO proteins at the membrane, for example, with a scaffolding protein (Figure 4) [137].

## 4. Downstream Effectors of RHO GTPases

The ability of RHO GTPases to control a wide range of intracellular signaling pathways is attributed to their association with their cellular targets: effector proteins (Figure 5, Table 1). In contrast to regulators that interact with RHO GTPases to modulate their switch function, effectors require GTPases to be in a specific conformation to realize their own intrinsic function. To date, more than 70 potential effectors have been identified for RHOA, RAC1, and CDC42 [58].

The effector proteins are either kinases or scaffolding proteins (Figure 5, Table 1). Kinases form an important class of RHO effectors and result in downstream phosphorylation cascades. Different RHO-associated serine/threonine kinases such as PAK (p21-activated kinase), ROCK (RHO-associated coiled-coil kinase), CRIK (citron kinase), and PKN (protein kinase novel) interact with and are regulated by their partner GTPases [138,139,140]. Another group of effectors comprise scaffolding proteins, which probably form a framework for signaling cascades, especially through filamentous actin dynamics. IQ motif-containing GTPase-activating protein 1 (IQGAP1) [141], mammalian homolog of Drosophila diaphanous 1 (DIA1), Wiskott-Aldrich syndrome protein (WASP), and Rhotekin (RTKN) are the most extensively investigated effectors in this regard and facilitate complex formation in cells [142].

### 4.1. Structural Characteristics of RHO GTPase-effector Interactions

The crystal structures of the GTPase-binding domains (GBDs) of PKN and RHO kinase (ROCK) in complex with RHOA revealed that the domains, as predicted from their primary structure, form α-helical coiled coils that are arranged in an antiparallel and parallel fashion, respectively [143,144]. A 13-residue left-handed coiled coil in the C-terminal portion of the ROCK-GBD, which is considered the minimal sequence required for RHO-interacting motif activity, binds exclusively to the switch and α2 regions of RHOA. In contrast, the RHOA-PKN complex has two possible contact sites on RHOA [143]: contact site 1 consists of the α1, β2/β3, and α5 regions of RHOA, whereas contact site 2 overlaps remarkably well with the ROCK-binding site. The structures of CDC42 in complex with effector proteins containing a CDC42/RAC-interactive binding (CRIB) motif such as PAK1 and WASP, which have been determined mostly by NMR spectroscopy due to their high flexibility [145,146,147,148,149], have shown that the GBD in this class of effectors makes extensive contact with the surface of RHO GTPases. Specifically, GBD binds through its β-hairpin and C-terminal α-helix to the α1, switch I, and II regions and wraps around the α5 and β2 regions of the GTPase with its extended N-terminus, which encompasses the CRIB motif. The basic region of WASP, immediately upstream of the CRIB motif, has been shown to generate favorable electrostatic steering forces to unique glutamate residues in CDC42 (Glu49, Glu171, and Glu178) that control the accelerated WASP-CDC42 association reaction (Figure 2) [52,150]. This process is a prerequisite for WASP activation and a critical step in the temporal regulation and integration of WASP-mediated cellular responses (Figure 5).

Two other effectors, arfaptin and p67^phox^, have novel structures and contact sites on the GTPase [151,152]. Arfaptin forms an elongated crescent-shaped dimer with three helix coiled-coils that makes contact with the switches I and II and α2 regions of RAC1, regardless of its nucleotide-bound state [152], and structurally mimics the DH domain of Tiam1 [153]. p67^phox^ has an α-helical domain that consists of four tetratricopeptide repeat (TPR) motifs, which bind α1, the N-terminal residues of switch I, and the G3 and G5 loops, but not the switch II region or the principal parts of switch I [151]. It has been proposed that the switch regions might be the contact sites for a third protein that is associated with the Rac1-GTP-p67^phox^ complex [154,155].

The mechanism of effector activation of the GTPase–effector complex structures mentioned thus far have not been clarified; however, intramolecular autoinhibition and exposure of their functional domains are known to be required. A common feature of effector complexes is that, with the exception of p67^phox^, they all make intensive contact with the switch/α2 regions of RHO GTPases, which indicates that this region probably serves as the platform for the GTP-dependent recognition of effectors. Two invariant leucine residues (Leu69 and Leu72), which form crucial hydrophobic contacts with almost all effector domains, have been proposed as essential elements for the CDC42/RAC-mediated activation of CRIB-containing effectors [148]. A different activation mechanism has been implicated for the RHO-specific effectors PKN and ROCK, with other domains that bind cooperatively to sites outside the switch regions of RHOA [156].

### 4.2. RHO GTPase-Mediated Effector Signaling

The fact that effectors commonly contact distinct residues within the highly conserved switch I and II regions of RHO GTPases [3,5] strongly suggests that other domains bind cooperatively to sites outside the switch regions [3,156]. This possibility might explain the cellular specificity of RHO GTPase–effector interactions. Pioneering experiments by Alan Hall and colleagues showed that the reorganization of the actin cytoskeleton is regulated by proteins in the RHO family including CDC42, RAC1, and RHOA (Figure 5) [157]. CDC42 and RAC1 activation, in turn, activates the ARP2/3 complex indirectly via WASP and WAVE to induce branched actin filament networks and the formation of tight bundles of parallel filaments that form the core in filopodia and the formation of a network of diagonally oriented actin filaments that give rise to thin sheets of lamellipodia. RHOA activation leads to the activation of ROCK and DIA and the organization of actomyosin bundles into stress fibers as well as the formation of focal adhesions [1,5]. Coordination of the distinct roles of these GTPases is crucial for regulating cell migration, as demonstrated by wound closure in a fibroblast monolayer: CDC42 regulates cell polarity, RAC1 regulates the protrusion of lamellipodia at the leading edge, and RHO regulates the turnover of highly organized structures termed focal adhesions (reviewed in [7,158,159,160,161,162,163,164,165]).

Moreover, RHO GTPases control signal transduction pathways that influence gene expression including the serum response factor (SRF), nuclear factor κB (NFκB) transcription factor, c-JUN N-terminal kinase (JNK), and p38 mitogen-activated protein kinase pathways [19,166]. It has been reported that several enzyme activities can be altered by RHO GTPases. RAC1 can bind directly to p67^phox^, a component of the NADPH oxidase complex, and activate NADPH oxidase activity to generate reactive oxygen species (ROS) (Table 1) [167]. The BCR gene produces a 160 kDa product called p160^bcr^, which encompasses several distinct domains. p160 exhibited GAP activity toward RAC1, RAC2, and CDC42 GTPases. An early study has shown that BCR regulates RAC-mediated superoxide production by the NADPH-oxidase system of leukocytes [168]. 

## 5. Conclusions

Abnormal activation of RHO proteins has been shown to play a crucial role in cancer, infectious and cognitive disorders, and cardiovascular diseases. However, several studies must be performed to gain understanding into the complexity of RHO protein signaling. (i) The RHO family comprises 20 signaling proteins, of which only RHOA, RAC1, and CDC42 have been comprehensively studied thus far. The functions of the less-characterized members of this protein family await detailed investigation. (ii) Despite intensive research over the past two decades, the mechanisms by which RHOGDIs associate and extract RHO proteins from the membrane and the factors displacing the RHO protein from the complex with RHOGDI remain to be elucidated. (iii) A tremendous number of 20 RHO-regulating proteins (85 GEFs and 66 GAPs) exist in the human genome. How these regulators selectively recognize their RHO protein targets is not well understood, and the majority of GEFs and GAPs in humans remain uncharacterized. (iv) GDIs, GEFs, GAPs, and effectors, despite their structural diversity, share consensus binding sites within the switch I and II regions [3]. However, all these RHO-binding partners require contact with other regions, not their shared binding region, to realize their specificity for different RHO proteins. (v) A major challenge ahead, which has not been fully addressed thus far, will be gaining an understanding of the spatial temporal regulation of RHO GTPase activity and the interaction of RHO proteins with distinct downstream effectors. (vi) Most GEFs and GAPs need to be regulated and their activation is generally achieved through the release of autoinhibitory elements [92,120]. With a few exceptions [4], the operating principles of these autoregulatory mechanisms remain obscure. (vii) A better understanding of the specificity and the mode of action of the regulatory proteins as well as the selective recruitment and activation of effectors to specific subcellular sites is not only fundamentally important for understanding many aspects of RHO biology, but is also the master key to unlocking the identity of key targets useful in developing drugs against a variety of diseases caused by aberrant RHO protein functions. This regards the spatiotemporal features whose understanding is afflicted with major conceptual shortcomings. Future models should consider both the emerging principle of biomolecular condensates (or non-membrane-bound organelles) that are assembled in liquid-liquid phase separation [169,170] and the modulating principle of accessory proteins [171,172,173], which appears to safeguard the strength, efficiency, and specificity of signal transduction.

## Figures and Tables

**Figure 1 cells-10-01831-f001:**
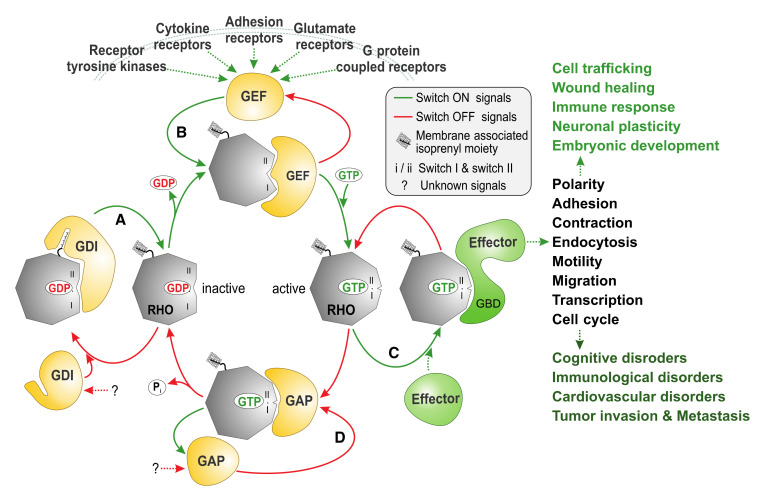
Molecular principles of RHO GTPase regulation and signaling. Most RHO GTPases (20 canonical human members) act as molecular switches by cycling between a GDP-bound, inactive state and a GTP-bound, active state. They interact specifically with four structurally and functionally unrelated classes of proteins: (**A**) In resting cells, guanine nucleotide dissociation inhibitors (GDIs; 4 human members) sequester RHO in the cytoplasm, away from the membrane, by binding to the lipid anchor and thus creating an inactivated cytosolic pool; (**B**) in stimulated cells, different classes of membrane receptors activate guanine nucleotide exchange factors (GEFs; 85 human members: 74 DBL and 11 DOCK family proteins), which in turn activate RHO by accelerating the intrinsic exchange of GDP for GTP and switch ON signal transduction; (**C**) active GTP-bound RHO interacts through the GTPase-binding domain (GBD) with and activates downstream targets (effectors; >70 human members) to cause a variety of intracellular pathways, which control a multitude of biochemical processes involved in the regulation of different biological (dys)functions; (**D**) GTPase-activating proteins (GAPs; 66 human members) negatively regulate RHO by stimulating its slow intrinsic GTP hydrolysis activity and switch OFF signal transduction. Notably, all RHO-interacting proteins recognize and bind RHO at consensus-binding sites called switch I and II.

**Figure 3 cells-10-01831-f003:**
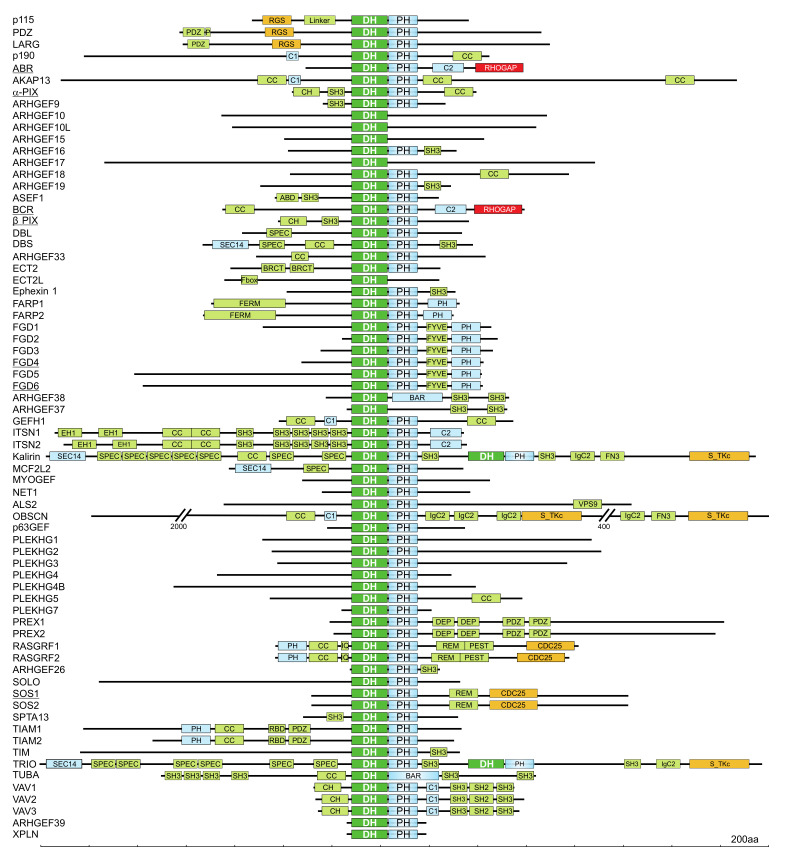
Domain organization of DBL family proteins. The DBL family RHOGEFs are mostly multimodular proteins and have a number of functional domains that may mediate cross talk between RHO proteins and other signaling pathways. DH domains are almost always found with a PH domain in the C-terminus. Some DBL proteins contain two DH–PH cassettes, while some DBL proteins lack tandem PH domains. Functional domains, in addition to the catalytic DH domain (green), are probably involved in lipid and membrane binding (blue), protein interactions (bright green), and enzymatic activities (red and orange). A scale of amino acid numbers in increments of 200 is shown at the bottom. Underlined proteins do not exhibit activity under cell-free conditions [47].

**Figure 4 cells-10-01831-f004:**
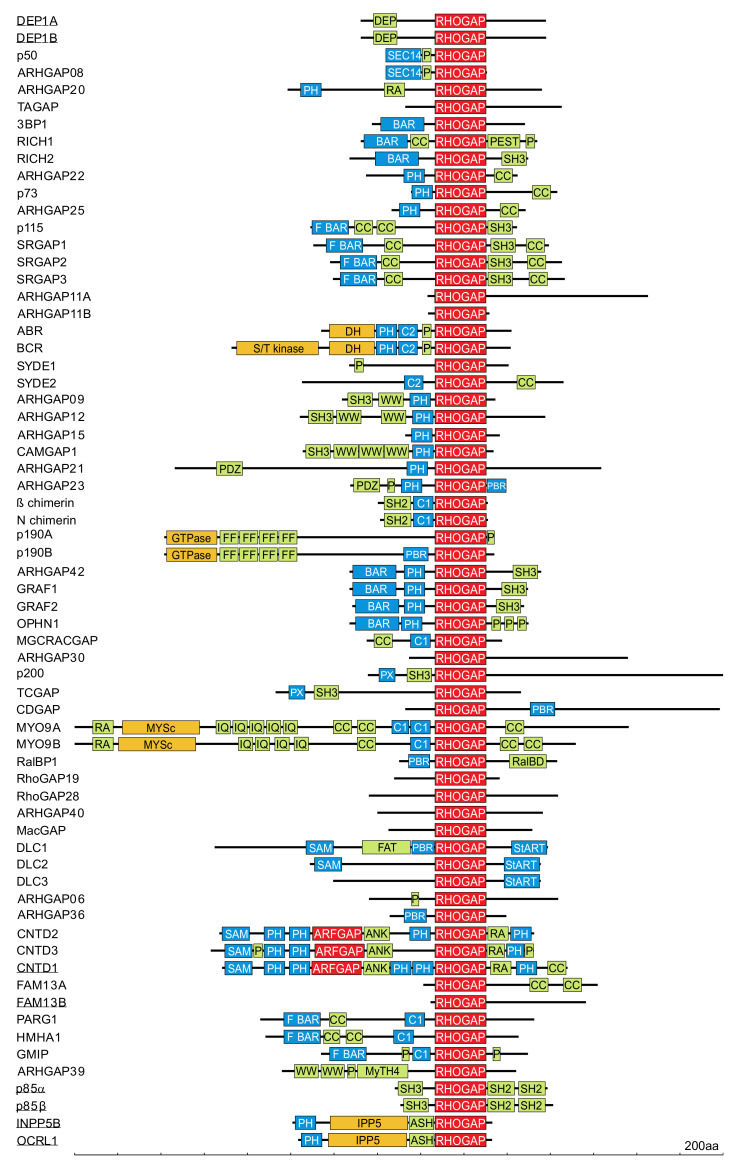
Domain organization of the RHOGAP family proteins (adapted from Amin et al., 2016 [114]. RHOGAPs are mostly multimodular proteins and have a number of functional domains that may mediate cross talk between RHO proteins and other signaling pathways. Functional domains, in addition to the catalytic GAP domain (red), are probably involved in lipid and membrane binding (blue), protein interactions (bright green), and enzymatic activities (red and orange). A scale of amino acid numbers in increments of 200 is shown at the bottom. Underlined proteins are GAP-like proteins with no RHOGAP activity [114].

**Figure 5 cells-10-01831-f005:**
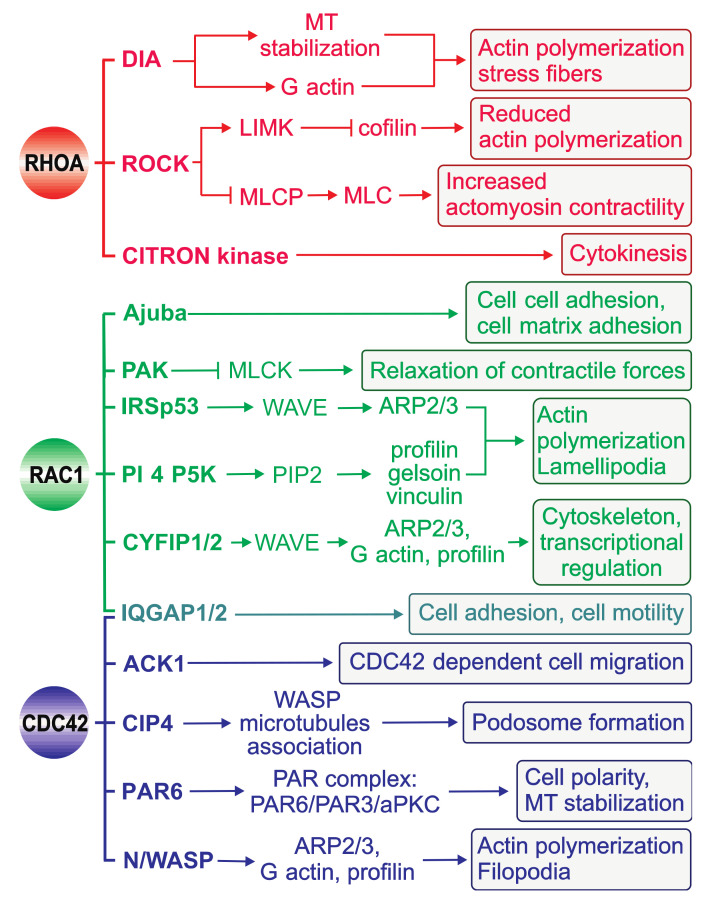
Regulation of actin-based motility by RHOA, RAC1, and CDC42. Activated CDC42, RAC, and RHO bind to and specifically activate their downstream effectors, which are either kinases (e.g., ROCK, PAK, and PI5K) or scaffolding proteins (e.g., DIA, WASP, IRSp53, and IQGAP). These effector proteins activate diverse signaling pathways with distinct effects on the actin cytoskeleton and cellular morphology. An important aspect of cell motility is the equilibrium between the myosin light chain (MLC) and phosphorylated MLC, which is tightly regulated.

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
