# Peer review of "The RHO Family GTPases: Mechanisms of Regulation and Signaling"

_cells, 2021, doi:10.3390/cells10071831_

Round 1
Reviewer 1 Report
This is a nice, complete, useful and up to date review on the RHO family regulation by Mosaddeghzadeh and Ahmadian. The figures are in general nice and useful and the overall depth and width of the review are adequate. I like to recommend acceptance after improving some smaller points listed below:
- The title promisses to talk on the regulation and signaling of Rho GTPases. In my opinion the main focus on text and figures (95%) is on the regulation. The signaling would have to include downstream effector functions and target molecules, which were not really covered here. I recommend to take out "signaling" from the title. Alternatively, another figure with the main down (and up-?) stream signaling partners of Rho (not their regulators!) could be included, and another additional page explaining potential downstream events.
2. figure 1: the green and yellow molecules are depicted as two diferent schematic versions, suggesting open/active and closed/inactive forms. However, the two versions are quite diferent and also the diferent molecule types (GEF vs. GDI vs. GAP etc...) differ in the way the conformational changes are depicted (in case of the GDI the open form seems to represent two and not one molecule!). I would suggest to opt for more schematic, uniform versions here, such as the changes depicted in fig.2 for Gefs, where the domains are represented by schematic boxes, rather, and there is no qualitative structural diferences between open/closed versions, but rather only presence or absecence of intra molecular interactions.
3. Figure 2: the authors opted to show the domain names/abreviations upside down in the inactive form , where ID contacts Effector domain (or ID contacting GEF/GAP). I think for the sake of readbility it would be better not to invert these !!! The figure itself is quite didactic and does not need this inversion of text.
Author Response
This is a nice, complete, useful and up to date review on the RHO family regulation by Mosaddeghzadeh and Ahmadian. The figures are in general nice and useful and the overall depth and width of the review are adequate. I like to recommend acceptance after improving some smaller points listed below:
1. The title promises to talk on the regulation and signaling of Rho GTPases. In my opinion the main focus on text and figures (95%) is on the regulation. The signaling would have to include downstream effector functions and target molecules, which were not really covered here. I recommend to take out "signaling" from the title. Alternatively, another figure with the main down (and up-?) stream signaling partners of Rho (not their regulators!) could be included, and another additional page explaining potential downstream events.
Authors’ response: We thank the reviewer for the constructive criticism, and totally agree that the raised expectations have not been fulfilled in the manuscript. This has now changed in the revised manuscript considerably. We have added a new Figure 5, and an expanded Table 1 about the effectors RHO, RAC1 and CDC42, and their downstream pathways, and described the different modes of GTPase-effector interactions.
2. Figure 1: the green and yellow molecules are depicted as two different schematic versions, suggesting open/active and closed/inactive forms. However, the two versions are quite different and also the different molecule types (GEF vs. GDI vs. GAP etc...) differ in the way the conformational changes are depicted (in case of the GDI the open form seems to represent two and not one molecule!). I would suggest to opt for more schematic, uniform versions here, such as the changes depicted in fig.2 for GEFs, where the domains are represented by schematic boxes, rather, and there is no qualitative structural differences between open/closed versions, but rather only presence or absence of intra molecular interactions.
Authors’ response: We thank the reviewer for these suggestions. It is true that closed/inactive forms can be misleading, especially in the case of the GDI. So, we changed these forms to simple shapes.
3. Figure 2: the authors opted to show the domain names/abbreviations upside down in the inactive form, where ID contacts Effector domain (or ID contacting GEF/GAP). I think for the sake of readability it would be better not to invert these!!! The figure itself is quite didactic and does not need this inversion of text.
Authors’ response: We thank the reviewer for bringing up this point. However, we removed Figure 2 from the manuscript as suggested by reviewer #3.
Reviewer 2 Report
This review summarizes the mechanisms of regulation of Rho GTPases.
It corresponds to a brief and general overview of the way Rho GTPases function. In the high number of reviews about Rho GTPases the add-value of this one is not obvious. Especially because most of the cited publications are pretty old, the more recent is from 2016. Major papers that appeared recently, such as Müller PM et al., NCB2020 or Bagci H et al., NCB2020 are not cited.
Here are the other major flaws and suggestions.
Suggested amendments
- Table I with Rho GTPase effectors is not described in the manuscript. Why only 10 over 20 GTPases are cited? What about RhoB, Rac2, RhoG, RhoH, …? A legend would be useful, what does mean the asterisk?
- The organization of the review is perfectible. DOCK family of RhoGEFs should be included in the 3.2 section. Some ideas appear twice at two different positions in the manuscript (ex: post-translational modification page 2 and page 4; description of DH and PH domains for GEFs).
- The involvement of DOCK proteins in diseases is particularly interesting. The dysregulation of DBL proteins is important to mention to describe them as “attractive therapeutic targets”
- The two figures are particularly well done. Other figures for the RhoGEF and RhoGAP sections would be valuable.
Minor corrections
- Abstract – the number of GDIs, GEFs and GAPs must be revised and consistent with the details in the review, i.e. 3 GDIs instead of 4, 66 or 68 RhoGAPs, number of GEFs versus number of DBL proteins…
- Abstract – “challenges and future perspectives are reviewed”
- MS – Rho (and not RHO), please be consistent through the whole manuscript.
Author Response
This review summarizes the mechanisms of regulation of Rho GTPases.
It corresponds to a brief and general overview of the way Rho GTPases function. In the high number of reviews about Rho GTPases the add-value of this one is not obvious. Especially because most of the cited publications are pretty old, the more recent is from 2016. Major papers that appeared recently, such as Müller PM et al., NCB2020 or Bagci H et al., NCB2020 are not cited.
Here are the other major flaws and suggestions.
Suggested amendments:
1. Table I with Rho GTPase effectors is not described in the manuscript. Why only 10 over 20 GTPases are cited? A legend would be useful, what does mean the asterisk?
Authors’ response: We thank the reviewer for raising critical points. We apologize for the insufficient descriptions at different parts of the manuscript, which we have completed in the revised manuscript.
1a. Table I with Rho GTPase effectors is not described in the manuscript.
We have described the content of Table 1 throughout the text.
1b. Why only 10 over 20 GTPases are cited? What about RhoB, Rac2, RhoG, RhoH, …?
Initially we decided for only two representatives of each subfamily. However, your suggestion convinced us to expanded the numbers of the effectors for other members of the RHO family in Table 1.
1c. A legend would be useful, what does mean the asterisk?
We added a legend for Table 1, and explained that “asterisk refers to the effectors, which are shared between RAC1 and CDC42.
2. The organization of the review is perfectible. DOCK family of RhoGEFs should be included in the 3.2 section. Some ideas appear twice at two different positions in the manuscript (ex: post-translational modification page 2 and page 4; description of DH and PH domains for GEFs).
Authors’ response: We thank the reviewer for raising these points. We have carefully check for missing points and redundancy throughout the manuscript and revised it adequately.
3. The involvement of DOCK proteins in diseases is particularly interesting. The dysregulation of DBL proteins is important to mention to describe them as “attractive therapeutic targets”
Authors’ response: We appreciate the reviewer´s suggestion. We included a paragraph in DOCK family of RHOGEFs and note the crucial role of these proteins in several diseases.
4. The two figures are particularly well done. Other figures for the RhoGEF and RhoGAP sections would be valuable.
Authors’ response: We thank the reviewer for the constructive comments. We added two new Figures, which illustrate the diversity and the domain organizations of the GEFs and GAPs.
Minor corrections
2. Abstract – the number of GDIs, GEFs and GAPs must be revised and consistent with the details in the review, i.e. 3 GDIs instead of 4, 66 or 68 RhoGAPs, number of GEFs versus number of DBL proteins…
Authors’ response: We thank the reviewer for the comments, and we apologize for the inconsistency. We have adjusted the numbers of the regulators throughout the manuscript, including the Abstract.
2. Abstract – “challenges and future perspectives are reviewed”
Authors’ response: We thank the reviewer for pointing out this mistake, which we corrected now. Moreover, we carefully check the manuscript for other possible mistakes, and obtained a professional editing of the revised manuscript by American Journal Experts.
3. MS – Rho (and not RHO), please be consistent through the whole manuscript.
Authors’ response: We thank the reviewer for this comment. According to an official nomenclature convention by the HUGO Gene Nomenclature Committee (HGNC, supported by National Human Genome Research Institute), human genes and gene products are written in capital letters; the gene names are written in italics and the gene products in non-italics. Therefore, we decided long time ago to follow this recommendation.
Reviewer 3 Report
The review by Mosaddeghzadeh and Ahmadian is a classical review about RhoGTPases interactors and regulators. It gives to the non-specialist a clear view of the subject, and in the same time to the specialist some interesting focus on structural data that are not frequently presented. I found the manuscript well written and up to date, and also historically documented. I nevertheless have some concerns that I hope will help the authors enhance the quality of the review.
- Concerning the figures, the first one is totally relevant, but the second concerns effectors and their regulation. The subject is not really treated in the text (a few lines) and do not necessitate a figure. In the same perspective, the table of effectors is certainly interesting (although incomplete, RhoG for example) but do not corresponds to any paragraph in the text. I suggest a development on effectors could be added to justify these figures and table, in particular in classifying these effectors by functions (which is a particularity of RhoGTPases: Kinases -same as RasGTPases- , but also actin polymerization regulators and so-called scaffolds -the term is problematic, WASP for example is not only a scaffold protein-).
- In the paragraph “Rho family and the molecular switch mechanism”, the subject is the switch mechanism, not the structure and localization of Rho proteins. In fact the insertion of some lines on the CAAX box mechanism seems not well positioned in the manuscript. I suggest displacing it in a more logical place. Moreover, the problem of localization of Rho GTPases is not really treated (different endomembranes and plasma membrane, relation to the presence of Farnesyl or Geranyl-geranyl), and a structural important feature totally lacks: the polybasic region sometimes replaced by phosphorylation sites at the C-terminal of Rho Proteins. This should be added.
- Concerning the GDIs, one information lacks: the presence of GDP or GTP-forms of GTPases in complex with GDI. This was the subject of different publications and should be added. Also, in the third paragraph, the expression “isoprenylation process” is problematic. We can’t decide if it is the processing of isoprenylation the C-ter of RhoGTPases, or the isoprenyl moiety. Here, I think moiety is more appropriated.
- The GEFs are very well treated (it is obviously the specialty of the authors) but some information lacks. First, there is no information about the activation of GEFs. I suggest to develop the pathways leading to RhoGTPases activation (in the same way that was done for GAPs). Also a table giving the specificities of GEFs for GTPases should be provided (The authors have already these informations). It would help integrating the subject.
- Moreover in the structure of GEFs, I found a mistaking sentence twice in page 6: “the DH domain is preceded by a PH domain.” This should be modified into The DH domain is followed by a PH domain, knowing that in all cases, this PH domain is C-terminal relative to DH domain (as noticed by the authors below). Also in paragraph 3.2.2, second phrase: The catalytic… reside entirely within (not with) the DH domain.
- Finally, in the GAPs paragraph, I suggest to develop the information about the importance of GAPs in RhoGTPase signaling by adding the information on Bcr KO that was published in Cell in 1995 by Voncken et al. In this paper Bcr-null mutants were unable to stop the NADPH activity of neutrophils stimulated by Gram- infection. This was due to their incapacity to stop the Rac1 activation pathway leading to superoxide overproduction.
After all these remarks, I shall note that this is an excellent review, and I enjoyed its reading.
Author Response
The review by Mosaddeghzadeh and Ahmadian is a classical review about RhoGTPases interactors and regulators. It gives to the non-specialist a clear view of the subject, and in the same time to the specialist some interesting focus on structural data that are not frequently presented. I found the manuscript well written and up to date, and also historically documented. I nevertheless have some concerns that I hope will help the authors enhance the quality of the review.
1. Concerning the figures, the first one is totally relevant, but the second concerns effectors and their regulation. The subject is not really treated in the text (a few lines) and do not necessitate a figure. In the same perspective, the table of effectors is certainly interesting (although incomplete, RhoG for example) but do not corresponds to any paragraph in the text. I suggest a development on effectors could be added to justify these figures and table, in particular in classifying these effectors by functions (which is a particularity of RhoGTPases: Kinases -same as RasGTPases- , but also actin polymerization regulators and so-called scaffolds -the term is problematic, WASP for example is not only a scaffold protein-).
Authors’ response: We thank the reviewer for raising critical points. We apologize for the insufficient descriptions at different parts of the manuscript, which we have completed in the revised manuscript. We expanded the effector section, Table 1 and included a new Figure that illustrates various signaling pathways.
1a. Concerning the figures, the first one is totally relevant, but the second concerns effectors and their regulation. The subject is not really treated in the text (a few lines) and do not necessitate a figure.
We thank the reviewer for pointing out this problem. Figure 2 should visualize the principle of autoregulation and emphasize the importance of the so-called autoinhibitory domain. Autoinhibition mechanism would create a reversible barrier, which hamper the spontaneous activation of signaling pathway, and enable the system to respond to right and specific signal. Therefore, we mentioned this matter throughout the article but agree to remove Figure 2 from the manuscript.
1b. In the same perspective, the table of effectors is certainly interesting (although incomplete, RhoG for example) but do not corresponds to any paragraph in the text.
We have expanded the numbers of the effectors in Table 1 for other members of the RHO family, such as RHOG. We referred to Table throughout the manuscript.
1c. I suggest a development on effectors could be added to justify these figures and table, in particular in classifying these effectors by functions (which is a particularity of RhoGTPases: Kinases -same as RasGTPases- , but also actin polymerization regulators and so-called scaffolds -the term is problematic, WASP for example is not only a scaffold protein-).
We thank the reviewer for the constructive comments. Therefore, we expanded the parts for the RHO GTPase signaling, by completing and justifying Table 1, and providing a new Figure describing different effector pathways.
2. In the paragraph “Rho family and the molecular switch mechanism”, the subject is the switch mechanism, not the structure and localization of Rho proteins. In fact the insertion of some lines on the CAAX box mechanism seems not well positioned in the manuscript. I suggest displacing it in a more logical place. Moreover, the problem of localization of Rho GTPases is not really treated (different endomembranes and plasma membrane, relation to the presence of Farnesyl or Geranyl-geranyl), and a structural important feature totally lacks: the polybasic region sometimes replaced by phosphorylation sites at the C-terminal of Rho Proteins. This should be added.
Authors’ response: We thank the reviewer for bringing up these points. Accordingly, we revised the manuscript by a clearer allocation of the RHO GTPases localization and mentioned also other posttranslational modifications, including phosphorylation, ubiquitylation, sumoylation, and acetylation. In this context, we included a new figure that illustrates different signatures of the RHO GTPases (see new Figure 2).
3. Concerning the GDIs, one information lacks: the presence of GDP or GTP-forms of GTPases in complex with GDI. This was the subject of different publications and should be added. Also, in the third paragraph, the expression “isoprenylation process” is problematic. We can’t decide if it is the processing of isoprenylation the C-ter of RhoGTPases, or the isoprenyl moiety. Here, I think moiety is more appropriated.
Authors’ response: We thank the reviewer for the very useful comment and agreed that “the isoprenyl moiety” fits much better than “isoprenylation process”. So, we changed this appropriately. Moreover, we expanded the GDI sections in the manuscript by referring to the nucleotide-dependent complex formation between GDIs and RHO GTPases.
4. The GEFs are very well treated (it is obviously the specialty of the authors) but some information lacks. First, there is no information about the activation of GEFs. I suggest to develop the pathways leading to RhoGTPases activation (in the same way that was done for GAPs). Also a table giving the specificities of GEFs for GTPases should be provided (The authors have already these informations). It would help integrating the subject.
Authors’ response: We thank the reviewer for this suggestion. There are excellent reviewer article covering this issue that are cited in our manuscript. However, we added information about the activation of various GEFs. The part about the specificities of GEFs is expanded, although we have referred to our JBC article (Jaiswal et al., 2013), which comprehensively discusses this issues.
5. Moreover in the structure of GEFs, I found a mistaking sentence twice in page 6: “the DH domain is preceded by a PH domain.” This should be modified into The DH domain is followed by a PH domain, knowing that in all cases, this PH domain is C-terminal relative to DH domain (as noticed by the authors below). Also in paragraph 3.2.2, second phrase: The catalytic… reside entirely within (not with) the DH domain.
Authors’ response: We thank the reviewer for pointing out this mistake, which is corrected in revised manuscript.
6. Finally, in the GAPs paragraph, I suggest to develop the information about the importance of GAPs in RhoGTPase signaling by adding the information on Bcr KO that was published in Cell in 1995 by Voncken et al. In this paper Bcr-null mutants were unable to stop the NADPH activity of neutrophils stimulated by Gram- infection. This was due to their incapacity to stop the Rac1 activation pathway leading to superoxide overproduction.
Authors’ response: We thank the reviewer for excellent suggestion. We have now included and described this study by Voncken et al. (1995) regarding the role of BCR in the regulation of RAC1.
7. After all these remarks, I shall note that this is an excellent review, and I enjoyed its reading.
Authors’ response: Thank you very much for reviewing our manuscript, and the valuable and insightful comments as well as outstanding ideas. We gave our best to respond to all comments, and took the ideas and implemented them into the revised manuscript, which in total have greatly helped to improve the overall quality of the fully revised manuscript.
Round 2
Reviewer 1 Report
This interesting and thoughtful review has improved a lot after revision. The authors incorporated all suggestions by the reviewers in an appropriate fashion. The new figures and tables are very informative and well composed.
Congrats! I now can fully recommend acceptance!
Author Response
We thank the reviewer for the feedback and we are happy the revisions were satisfactory.Reviewer 2 Report
The authors have provided a revised manuscript and have done a good job to answer the reviewers’ comments.
The new version of the review is complete, clear and well illustrated.
Author Response
We thank the reviewer for the feedback and we are happy the revisions were satisfactory.Reviewer 3 Report
I thank the authors for the revision of the manuscript. It is clearly improved and I appreciate the modifications done. The changes in the figures are accurate and give a better understanding of text descriptions.
I still have little concerns that will not require another reviewing process. In their response to point 6, the authors claim that they added some elements concerning Bcr, but I hardly can find it.
Also, in response to point 5, their assure to have changed the sentence "The DH domain is preceded by a PH domain": this is not the case, I found the phrase unchanged.
Finally, on line 106, there is a mention for fig 2 that does not correspond to anything in the figure. it should be removed (probably from the previous version).
After these modifications, I will not have any concerns.
Author Response
Date: July 12th, 2021
Manuscript ID: cells-1197167 - Minor Revisions
Dear Reviewer, thank you very much for your time and efforts. We have revised the manuscript regarding the minor changes you have suggested as follows:
1) I still have little concerns that will not require another reviewing process. In their response to point 6, the authors claim that they added some elements concerning Bcr, but I hardly can find it.
- Authors’ response: We have included on page 11 the following sentence and have cited Voncken et al., cell 1995, as suggested by the reviewer #3.
2) Also, in response to point 5, their assure to have changed the sentence "The DH domain is preceded by a PH domain": this is not the case, I found the phrase unchanged.
- Authors’ response: We changed the sentence "The DH domain is preceded by a PH domain" for "The DH domain is followed by a PH domain", as suggested by the reviewer #3.
3) Finally, on line 106, there is a mention for fig 2 that does not correspond to anything in the figure. it should be removed (probably from the previous version). After these modifications, I will not have any concerns.
- Authors’ response: We omitted (Fig. 2) in this sentence “on average, two orders of magnitude faster than GDP/GTP exchange” on page 3.